# Exploring PCSK9 Genetic Impact on Lipoprotein(a) via Dual Approaches: Association and Mendelian Randomization

**DOI:** 10.3390/ijms241914668

**Published:** 2023-09-28

**Authors:** Ya-Ching Chang, Lung-An Hsu, Yu-Lin Ko

**Affiliations:** 1Department of Dermatology, Chang Gung Memorial Hospital, College of Medicine, Chang Gung University, Tao-Yuan 33305, Taiwan; ycchang@cgmh.org.tw; 2Cardiovascular Division, Chang Gung Memorial Hospital, College of Medicine, Chang Gung University, Tao-Yuan 33305, Taiwan; 3Department of Research, Division of Cardiology, Department of Internal Medicine, Taipei Tzu Chi Hospital, Buddhist Tzu Chi Medical Foundation, New Taipei 23142, Taiwan; yulinkotw@yahoo.com.tw

**Keywords:** *PCSK9* gene, lipoprotein(a), polymorphism, Mendelian randomization, dyslipidemia, hyperlipoproteinemia(a)

## Abstract

Previous investigations have suggested an association between the *PCSK9* common polymorphism E670G and Lipoprotein(a) (Lp(a)) levels, as well as a link between plasma PCSK9 levels and Lp(a) concentrations. However, the causal relationship between plasma PCSK9 and Lp(a) levels remains uncertain. In this study, we explored the association between *PCSK9* E670G polymorphism and Lp(a) levels in 614 healthy Taiwanese individuals. Employing a two-sample Mendelian randomization (MR) analysis using openly accessible PCSK9 and Lp(a) summary statistics from the genome-wide association studies (GWAS) and UK Biobank, we aimed to determine if a causal link exists between plasma PCSK9 levels and Lp(a) concentrations. Our findings reveal that the E670G G allele is independently associated with a decreased likelihood of developing elevated Lp(a) levels. This association persists even after adjusting for common cardiovascular risk factors and irrespective of lipid profile variations. The MR analysis, utilizing six *PCSK9* GWAS-associated variants as instrumental variables to predict plasma PCSK9 levels, provides compelling evidence of a causal relationship between plasma PCSK9 levels and Lp(a) concentration. In conclusion, our study not only replicates the association between the *PCSK9* E670G polymorphism and Lp(a) levels but also confirms a causative relationship between PCSK9 levels and Lp(a) concentrations through MR analysis.

## 1. Introduction

The *PCSK9* gene, encoding proprotein convertase subtilisin/kexin type 9, has emerged as a critical player in lipid metabolism and cardiovascular health. The PCSK9 protein interacts with the low-density lipoprotein receptor (LDLR) and regulates its degradation, subsequently influencing the levels of circulating LDL cholesterol (LDL-C) [1]. The *PCSK9* gene’s intricate role in lipid metabolism and its influence on cardiovascular health have garnered significant attention. With its well-established impact on LDL-C regulation, PCSK9 has taken center stage in the pursuit of novel therapeutic avenues [2].

Elevated lipoprotein(a) (Lp(a)) levels independently contribute to atherosclerotic cardiovascular diseases through mechanisms involving atherogenesis, inflammation, and thrombosis [3]. Lp(a) is a distinctive lipoprotein characterized by its LDL core, which includes apolipoprotein B-100 (apoB-100) and its unique feature of covalently binding apolipoprotein(a) (apo(a)) [4]. This dual composition of apoB-100 and apo(a) sets Lp(a) apart. Despite containing LDL-like components, Lp(a) maintains its significance as a potent risk factor even when LDL cholesterol levels are lowered. However, lacking standardized assays, universal guidelines, and targeted treatments hinders management [3]. Given that Lp(a) levels are predominantly influenced by genetics, accounting for 70–90% of the variability [3], it is crucial to comprehend the genetic and biological factors that contribute to the variation in Lp(a) levels across different ancestries.

Despite the effectiveness of statins in increasing LDLR expression and lowering LDL cholesterol, they have exhibited limited success in reducing Lp(a) levels [5]. In contrast, PCSK9 inhibitors have been shown in recent investigations to reduce Lp(a) concentrations by approximately 25–30% [6], which underscores the connection between PCSK9 and Lp(a) levels. Furthermore, *PCSK9* E670G polymorphism has been associated with variations in PCSK9 function (gain or loss of function), as documented in the previous studies [7,8,9,10], in addition to its influence on LDL cholesterol levels. It is suggested that this genetic variant might affect the binding of PCSK9 to LDLR or other cellular interactions, potentially leading to changes in the LDLR-mediated clearance of lipoproteins, including Lp(a) [10]. Building upon the initial demonstration of the link between the *PCSK9* E670G polymorphism and Lp(a) levels, we further sought to extend our understanding by employing a Mendelian randomization (MR) approach. MR is a crucial analytical approach, offering a robust framework to investigate causal relationships between specific genetic variants, biomarkers, and clinical outcomes [11]. MR leverages the principles of Mendel’s laws of inheritance, utilizing genetic variants as instrumental variables to mimic a randomized controlled trial setting. This allows us to explore the causal effects of exposures such as plasma PCSK9 level on clinical outcomes such as Lp(a) concentrations using *PCSK9* genetic variants as proxies. By relying on naturally occurring genetic variation, MR mitigates issues of reverse causality and confounding that often challenge traditional observational studies. Through this method, we aimed to unveil a causal relationship between PCSK9 levels and Lp(a) concentrations.

## 2. Results

### 2.1. Hyperlipoproteinemia(a) and Plasma Levels of Lipid

A total of 614 subjects were enrolled in the study, recruited during routine health examinations. Table 1 lists the baseline characteristics of the study subjects. The study cohort was stratified into three groups based on Lp(a) levels: the low-risk group, defined by Lp(a) levels <9.4 mg/dL; the average-risk group, defined by Lp(a) levels ranging from 9.4 to 30 mg/dL; and the high-risk group, categorized by Lp(a) levels exceeding 30 mg/dL. An Lp(a) level exceeding 30 mg/dL commonly defines hyperlipoproteinemia(a) [12]. In our healthy population, 19% of individuals exhibit levels greater than 30 mg/dL, whereas in other ethnic populations, both with and without cardiovascular disease, this range varies from 7% to 35% [12]. Among the three groups, there were no significant differences detected in terms of age, body mass index (BMI), sex distribution, prevalence of hypertension, diabetes mellitus, and smoking frequency. Total cholesterol and LDL-C levels displayed a progressive and significant elevation corresponding to the hyperlipoproteinemia(a) risk categories. However, there were no discernible differences in high-density lipoprotein cholesterol (HDL-C) and triglyceride levels across the three groups.

### 2.2. PCSK9 Genotypes and Plasma Levels of Lipid

As demonstrated in our preceding study [13], the *PCSK9* E670G (c.2009A>G) G allele frequency was 6.0%, with genotype frequencies in Hardy–Weinberg equilibrium. Only one female subject exhibited *PCSK9* 670G allele homozygosity, showing an LDL-C level of 97 mg/dL. Consequently, the AG heterozygous and GG homozygous subjects were grouped together for further analysis. Analytical results indicated a significant link between the *PCSK9* 670G allele and lower LDL-C levels (*p* = 0.022). Carriers of the *PCSK9* 670G allele (AG or GG) had notably lower LDL-C levels (107 ± 32 mg/dL) compared to AA homozygous subjects (117 ± 33 mg/dL). Multiple linear regression confirmed *PCSK9* E670G as an independent predictor of LDL-C levels, accounting for age, gender, smoking, hypertension, diabetes mellitus, BMI, and lipid-lowering agent use (*p* = 0.029).

### 2.3. PCSK9 Genotypes and Plasma Levels of Lp(a)

Table 2 presents the *PCSK9* E670G genotype and the allele frequencies of the study participants categorized by hyperlipoproteinemia(a) risk levels. Notably, individuals carrying the *PCSK9* 670G allele exhibited notably diminished hyperlipoproteinemia(a) frequencies, showcasing a dose-dependent relationship (*p* = 0.010). Specifically, AG and GG genotypes displayed the highest frequencies associated with low-risk hyperlipoproteinemia(a) (*p* = 0.011). Using a multinomial logistic regression analysis, we investigated the association between the E670G G allele and the risk of hyperlipoproteinemia(a), considering different risk categories. Our findings revealed that individuals carrying the E670G G allele had significantly reduced probabilities of experiencing average (OR = 0.50, 95% CI = 0.29–0.87, *p* = 0.015) and high (OR = 0.40, 95% CI = 0.19–0.86, *p* = 0.019) risks associated with hyperlipoproteinemia(a) (Table 3). These results suggest that individuals with the G allele have a significantly lower risk of developing hyperlipoproteinemia(a), with a 50% lower chance for those at average risk and a 60% lower chance for those at high risk, compared to individuals with the AA genotype. This association remains significant even after adjusting for common cardiovascular risk factors, including age, gender, hypertension, smoking, diabetes mellitus, and BMI (Table 3). Furthermore, to account for the potential influence of lipid profiles, we conducted additional analyses by including cholesterol, LDL-C, HDL-C, and triglyceride levels in the model. Importantly, these adjustments did not alter the observed associations (OR = 0.53, 95% CI = 0.30–0.93, *p* = 0.026 for average risk; OR = 0.46, 95% CI = 0.21–0.99, *p* = 0.047 for high risk; as shown in Table 3). These results highlight the robustness of our findings and suggest that the E670G G allele is independently associated with a decreased risk of hyperlipoproteinemia(a), irrespective of lipid profile variations.

### 2.4. MR Study

To explore the potential causal relationship between plasma PCSK9 levels and Lp(a) concentrations, we conducted a two-sample MR study utilizing the MR-based platform. The previous genome-wide association studies (GWAS) consistently highlighted the *PCSK9* gene locus as strongly associated with plasma PCSK9 levels [14,15,16]. Accordingly, we meticulously selected six independent single nucleotide polymorphisms (SNPs) within the *PCSK9* gene as instrumental variables (IVs). Our SNP choices adhered to stringent criteria, including significant associations with PCSK9 levels that were consistently replicated in two independent GWAS studies. These associations yielded *p*-values reaching or exceeding the genome-wide significance threshold of 5.00 × 10^−7^. Moreover, we ensured that the pairwise linkage disequilibrium (LD) r^2^ values between these SNPs were all below 0.5, as outlined in the methods section. These chosen SNPs are rs11206510, rs11591147, rs2479409, rs45448095, rs499718, and rs557435 (as shown in Table 4). Remarkably, these SNPs exhibited noteworthy correlations with plasma PCSK9 levels, reaching or nearing the genome-wide significance threshold (with *p*-values ranging from 2.22 × 10^−7^ to 1.94 × 10^−17^, as depicted in Table 4). Collectively, these genetic variants explained 6% of the variance (R^2^) in plasma PCSK9 levels. Our two-sample MR analysis results unveiled a genetic predisposition indicating that elevated plasma PCSK9 levels are associated with a significant increase in inverse-rank normal transformed (irnt) Lp(a) levels (Table 5, Figure 1). This relationship is supported by the beta coefficients (β) obtained through different MR methods, including the inverse-variance weighted (IVW), weighted median, and MR-Egger analysis (Table 5). Specifically, in the IVW analysis, each unit increase in plasma PCSK9 levels corresponds to a 0.06933 unit increase in irnt Lp(a) levels (SE = 0.02007, *p* = 0.0005528). This suggests a robust and statistically significant positive association between PCSK9 and Lp(a). The weighted median analysis also confirms this association, with a beta coefficient of 0.06628 (SE = 0.02888, *p* = 0.02176). Furthermore, the MR-Egger analysis, which accounts for potential pleiotropy, also indicates a positive relationship, with a beta coefficient of 0.1083 (SE = 0.0383, *p* = 0.04741). The consistent findings across different MR methods strengthen our confidence in the observed association. Notably, the MR-Egger regression analysis revealed no evidence of directional pleiotropy effects (intercept = −0.034, *p* = 0.299). As shown in Figure 1b, the scatterplot of the effect sizes of the associations between the PCSK9-level-determining alleles and PCSK9 levels versus the effect sizes of the association between the PCSK9-level-determining alleles and Lp(a) concentrations in the MR analysis revealed that independent genetic variants in the *PCSK9* gene region were concordantly associated with the outcomes, supporting a causal relationship between PCSK9 levels and Lp(a) concentrations. These results provide compelling evidence that genetically determined higher plasma PCSK9 levels causally contribute to elevated Lp(a) levels. Furthermore, Cochran’s Q test did not identify significant heterogeneity (Table 5). Figure 2 displays the matrix of pairwise linkage disequilibrium statistics among the *PCSK9* E670G (rs505151) and six other PCSK9-level-determining SNPs.

## 3. Discussion

In this article, we present a strong association between the *PCSK9* E670G polymorphism and Lp(a) levels in a healthy population. Employing a two-sample MR analysis with the *PCSK9* genetic instrument, our study also provides evidence of a causal relationship between plasma PCSK9 levels and Lp(a) concentrations.

Previous studies have underscored the significance of *PCSK9* E670G polymorphism and its associated haplotype in influencing LDL cholesterol levels and the severity of coronary atherosclerosis [8,10]. Recent findings by Cui et al. indicate that *PCSK9* E670G AG + GG genotypes among patients on maintenance hemodialysis with atherosclerosis exhibit elevated Lp(a) levels compared to the AA genotype [17]. Nevertheless, our research paints a different picture, revealing that the G allele of the E670G variant is associated with lower, not higher, LDL-C and Lp(a) levels. This apparent paradox may stem from the likely non-functional nature of the *PCSK9* E670G polymorphism, leading to varied effects on LDL-C and Lp(a) levels across diverse study populations [7,9,12]. This variation could be due to its linkage disequilibrium with other functional mutations within or near the *PCSK9* gene [9]. Interestingly, a GWAS focusing on Lp(a) concentration has identified a risk locus of significance, marked by *PCSK9* rs11591147 (R46L) [18]. Although the *PCSK9* R46L variant is absent in the Taiwanese population, our prior regional plot association study, accompanied by conditional analysis of the *PCSK9* locus within the Taiwan Biobank, revealed that *PCSK9* E670G displayed genome-wide significant associations with LDL-C levels [9]. These suggest that *PCSK9* gene variants might serve as potential proxies not only for LDL-C levels but also for Lp(a) levels. Furthermore, prior investigations have highlighted an independent, positive relationship between serum PCSK9 and Lp(a) levels [19,20], suggesting a causal connection between the two. However, the limitations inherent in observational studies, including residual confounding and reverse causation, preclude definitive conclusions regarding plasma PCSK9’s impact on Lp(a). Through robust MR analysis, we successfully validate the causal association, offering insight into the intricate interplay between PCSK9, Lp(a), and cardiovascular health. Importantly, our findings indicate that the link between *PCSK9* E670G and Lp(a) is independent of cholesterol and LDL-C levels, reinforcing the potential of PCSK9 inhibitors to reduce Lp(a) levels effectively.

Our MR outcomes align with the notion that augmented PCSK9 expression may facilitate LDLR clearance, indirectly impacting Lp(a) clearance due to shared structural features [21]. Monoclonal antibodies targeting PCSK9 have demonstrated the ability to reduce Lp(a) by around 25–30% [22], and even the use of inclisiran, an siRNA against PCSK9 synthesis, showcases similar efficacy in reducing Lp(a) levels [23]. However, despite statins’ ability to increase LDLR expression, they do not substantially decrease Lp(a) levels. This implies that alternative pathways, possibly involving PCSK9’s interaction with distinct receptors for Lp(a) clearance, such as LDL receptor-related protein 1 (LRP1), cluster of differentiation 36 receptor (CD36), toll-like receptor 2 (TLR2), scavenger receptor-B1 (SR-B1), and plasminogen receptors, or even with the synthesis, assembly, and secretion of apolipoprotein(a), contribute to Lp(a) regulation [4,22]. Chemello et al.’s work indicated that altering LDLR expression with alirocumab did not impact the cellular or the hepatic uptake of Lp(a), suggesting that the LDL receptor is not a primary avenue for Lp(a) plasma clearance [24]. The intricate mechanisms governing the effect of PCSK9 inhibition on Lp(a) metabolism remain complex and not entirely understood, warranting further investigation. Based on our findings, treatment with PCSK9 inhibitors for hypercholesterolemia offers an additional benefit of lowering Lp(a) levels. This is particularly advantageous for individuals at high risk for atherosclerotic cardiovascular disease and elevated lipoprotein(a). Lowering PCSK9 levels, possibly through the use of combined different kinds of PCSK9 inhibitors, appears to enhance the control of Lp(a) levels and may lead to improved cardiovascular outcomes.

However, our study has limitations. First, the genetic association study involves a relatively small sample size. The post hoc statistical power analysis, assuming an odds ratio of approximately 0.5 for G allele carriers in terms of a 38% prevalence of average-risk or a 19% prevalence of high-risk hyperlipoproteinemia(a), indicates that this study’s statistical power, given the sample size of 614 participants and the 11.9% incidence of AG and GG genotypes, is only 68% and 40%, respectively, at a significance level of 0.05. While our study provides valuable insights, it is clear that a larger sample size would be essential to validate and strengthen our findings. Moreover, both the genetic association study conducted in the Taiwanese population and the MR study performed in major European ancestry populations have unveiled ethnic genetic heterogeneity. As a result, the implications of our findings might not extend to other ethnic groups. Additionally, the absence of functional assays and plasma PCSK9 levels in our study population hindered the exploration of whether the *PCSK9* genetic variant (E670G) affects PCSK9’s interaction with LDLR or other cellular processes, influencing LDLR-mediated Lp(a) clearance. Furthermore, our MR study relies on publicly available summary statistics, limiting our ability to adjust for potential confounders such as age, gender, smoking, LDL-C levels, and the use of lipid-lowering agents such as statins. Finally, the absence of mechanistic investigations to elucidate the underlying pathways and mechanisms by which the *PCSK9* gene controls Lp(a) levels is a notable limitation. As a result, caution should be exercised when attempting to generalize or extrapolate our study results to clinical science and practice.

## 4. Materials and Methods

### 4.1. Study Population

A total of 614 subjects were enrolled in the study. The subjects were recruited during routine health examinations between August 2003 and September 2004. The presence of hypertension, diabetes mellitus, hypercholesterolemia, or smoking was determined based on history, previous medical records, and current medication. Obesity was defined by a BMI of 25 kg/m^2^ or higher. The study protocol was approved by the Ethics Committee of Chang Gung Memorial Hospital, and informed consent was obtained from all subjects.

### 4.2. Blood Chemistry

A 10-mL blood sample was obtained from each subject following 12–14 h of overnight fasting. The Lp(a) concentration was measured by the latex-enhanced turbidimetric assay (Daiichi Pure Chemicals Co., Ltd., Tokyo, Japan). The total cholesterol, triglyceride, and glucose levels were measured using enzymatic methods (cholesterol oxidase/peroxidase-aminoantipyrine for cholesterol levels, glycerol phosphate oxidase/peroxidase-aminophenazone for triglyceride levels, and hexokinase method for glucose levels) on a Hitachi 7600-210 autoanalyzer (Hitachi, Tokyo, Japan) using commercially available kits. The levels of HDL-C were measured enzymatically following phosphotungsten/magnesium precipitation, and the LDL-C levels were calculated using the Friedewald formula {LDL cholesterols = total cholesterol–HDL cholesterol–(triglycerides/5)}. All of the measurements were performed in the Central Laboratory of the Department of Clinical Pathology of our hospital, and strict quality control procedures were followed. The inter-assay and intra-assay coefficients of variation were ~2% for the measurements.

### 4.3. Genomic DNA Extraction and Genotyping

The DNA was extracted using the Puregene DNA Isolation Kit (Gentra Systems, Minneapolis, MN, USA). Genotyping for the *PCSK9* gene’s E670G polymorphism was conducted through polymerase chain reaction (PCR) and restriction fragment length polymorphism, as previously described [13]. The amplification of exon 12 involved forward and reverse primers, resulting in a 386-bp product. The PCR products were then treated with MboII enzyme, generating two fragments (234 and 152 bp) in wild-type samples but not in those with the E670G polymorphism. To validate the genotyping results, a subset of samples was re-genotyped by a direct sequence analysis using the same primers employed for PCR amplification, yielding consistent outcomes.

### 4.4. MR Analysis

We conducted an MR analysis by using the database and analytical platform *MR-Base* [25] (http://www.mrbase.org/; App version: 1.2.2 3a435d). When the summary statistics on the exposure and outcome were acquired from independent GWAS, two-sample MR was used to search the causal effects. Three different estimate methods were used: inverse-variance weighted (IVW), MR-Egger regression, and weighted median [25]. Six independent *PCSK9* variants displaying pairwise LD r^2^ values below 0.5 were selected as IVs to represent plasma PCSK9 levels as the exposure. These choices were made based on their noteworthy associations, each exceeding the significance threshold of 5.00 × 10^−7^, as replicated in two independent GWAS examining plasma PCSK9 levels [14,15]. As the exposure, we used openly accessible summary statistics from Pott’s GWAS [14] for plasma PCSK9 levels from the 3290 people. As the outcome, we used summary statistics from a Lp(a) GWAS in the UK Biobank (ukb-d-30790_irnt), which included 27,386 people [25].

### 4.5. Statistical Analysis

An x^2^-test was used to examine differences among categorical variables. The clinical characteristics of continuous variables were expressed as mean ± SD and tested using a two-sample *t*-test or analysis of variance (ANOVA). A generalized linear model was used to analyze the LDL-C levels regarding the investigated genotypes and confounders. A multinominal logistic regression analysis was used to assess the independent effect of the investigated genotypes on the risks of hyperlipoproteinemia(a), with average risk defined as Lp(a) levels 9.4~30 mg/dL and high risk defined as Lp(a) levels >30 mg/dL, after controlling for the presence of known risk factors. Triglycerides were logarithmically transformed before statistical analysis to meet a normality assumption. LD was calculated using the LDmatrix software (https://analysistools.nci.nih.gov/LDlink/?tab=ldmatrix, version LDlink 5.6.5, (accessed on 25 August 2023)). All of the statistical analyses were performed using SPSS (version 22; SPSS, Chicago, IL, USA). The post hoc power analysis was calculated with G*Power 3.1.7 (http://www.psycho.uni-duesseldorf.de/abteilungen/aap/gpower3, (accessed on 21 September 2023)).

## 5. Conclusions

Our study replicates the observed link between the *PCSK9* E670G polymorphism and Lp(a) levels. It is important to note the limitation of a relatively small sample size and the results of our post hoc power analysis. Our MR analysis further supports a causative relationship between PCSK9 levels and Lp(a) concentrations. However, it is crucial to emphasize that validation from a larger, more diverse population and mechanistic investigations are necessary to enhance the robustness of our findings. Despite this limitation, our research significantly contributes to our understanding of the genetic and molecular factors underlying cardiovascular risk and offers promising avenues for potential therapeutic interventions. Further investigations with larger sample sizes are warranted to confirm and build upon our discoveries.

## Figures and Tables

**Figure 1 ijms-24-14668-f001:**
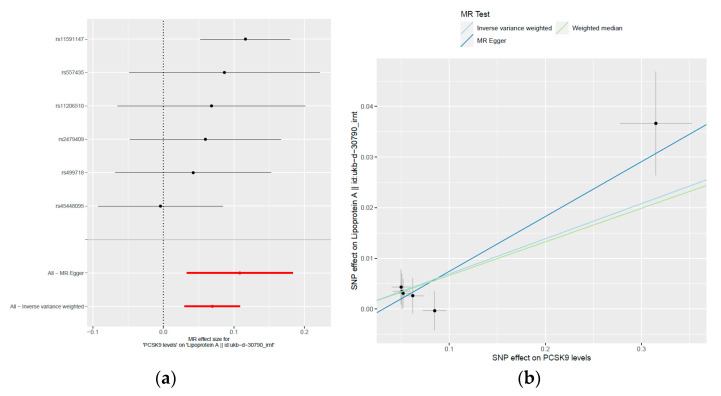
(**a**) Forest plot of causality of the single nucleotide polymorphisms associated with plasma PCSK9 levels on Lp(a) concentrations; (**b**) Scatter plot of genetic associations with plasma PCSK9 levels versus genetic associations with Lp(a) concentrations. Each method’s causal association is shown by the slopes of each line. The inverse-variance weighted estimate is shown in blue, the weighted median estimate is shown in green, and the Mendelian randomization Egger estimate is shown in dark blue. SNP, single nucleotide polymorphism; irnt, inverse-rank normal transformed.

**Figure 2 ijms-24-14668-f002:**
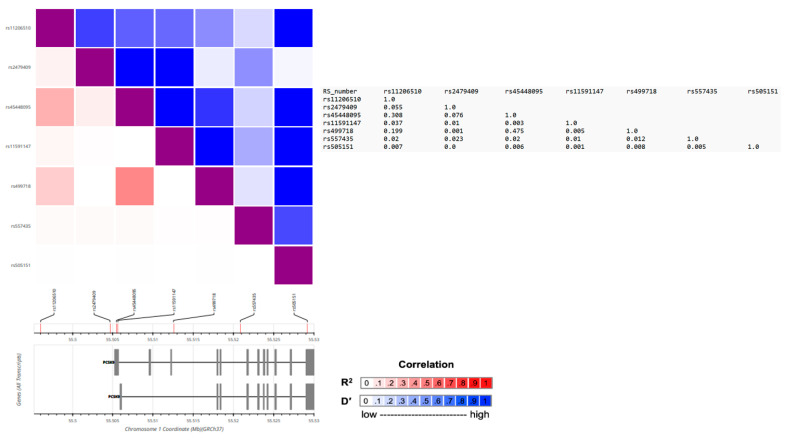
Location and linkage disequilibrium of the *PCSK9* E670G (rs505151) and six other PCSK9-level-determining variants within the European population. The varying shades of blue and red indicate the strength of pairwise linkage disequilibrium based on D’ and r^2^ values, with the corresponding r^2^ values displayed on the right side.

**Table 1 ijms-24-14668-t001:** Baseline characteristics of the study population according to Lipoprotein(a) levels.

	Lp(a) < 9.4 mg/dL	Lp(a) 9.4~30 mg/dL	Lp(a) > 30 mg/dL	*p*
	(n = 262)	(n = 233)	(n = 119)	
Age, years	45 ± 10	47 ± 11	47 ± 10	0.078
Male, (%)	54.2	53.6	52.5	0.953
Smoker, (%)	25.6	25.5	21.7	0.676
Hypertension, (%)	11.8	13.2	10.8	0.795
Diabetes, (%)	1.9	2.6	4.2	0.435
Body mass index, kg/m^2^	24.5 ± 3.7	24.3 ± 3.3	24.0 ± 3.3	0.488
Cholesterol, mg/dL	192 ± 36	199 ± 37	207 ± 37	0.001
LDL-cholesterol, mg/dL	109 ± 30	117 ± 33	127 ± 35	<0.001
HDL-cholesterol, mg/dL	55 ± 15	55 ± 15	55 ± 13	0.938
Triglyceride, mg/dL	153 ± 148	139 ± 91	124 ± 83	0.269

Triglycerides were logarithmically transformed before statistical testing to meet the assumption of normal distribution; however, untransformed data are shown.

**Table 2 ijms-24-14668-t002:** Distribution of genotypes and alleles for *PCSK9* E670G (c.2009A>G) according to Lipoprotein(a) levels.

	Lp(a) < 9.4 mg/dL	Lp(a) 9.4~30 mg/dL	Lp(a) > 30 mg/dL	Total	*p*
	(n = 262)	(n = 233)	(n = 119)	(n = 614)	
Genotype					0.044
AA	219 (83.6%)	212 (91.0%)	110 (92.4%)	541 (88.1%)	
AG	42 (16.0%)	21 (9.0%)	9 (7.6%)	72 (11.7%)	
GG	1 (0.4%)			1 (0.2%)	
AG+GG vs. AA	43 (16.4%)	21 (9.0%)	9 (7.6%)		0.011
Allele					
G, %	8.4	4.5	3.8	6.0	0.010

**Table 3 ijms-24-14668-t003:** Logistic regression analysis of the association between *PCSK9* E670G (c.2009A>G) genotypes and risk of hyperlipoproteinemia(a).

	Average Risk		Highest Risk	
	OR (95%)	*p*	OR (95%)	*p*
Model 1				
*PCSK9* E670G AG+GG vs. AA	0.50 (0.29–0.87)	0.015	0.40 (0.19–0.86)	0.019
Model 2				
*PCSK9* E670G AG+GG vs. AA	0.53 (0.30–0.93)	0.026	0.46 (0.21–0.99)	0.047

Risk of hyperlipoproteinemia(a): Reference, Lp(a) <9.4 mg/dL; Average risk, Lp(a) 9.4 ~ 30 mg/dL, High risk, Lp(a) >30 mg/dL. Model 1 variables included age, gender, smoking, body mass index, hypertension, diabetes mellitus, and *PCSK9* E670G. Model 2 variables included age, gender, smoking, body mass index, hypertension, diabetes mellitus, *PCSK9* E670G, and lipid profile (cholesterol, LDL-C, HDL-C, and triglycerides). OR, odds ratio; CI, confidence interval.

**Table 4 ijms-24-14668-t004:** Instrumental SNPs associated with PCSK9 and lipoprotein(a) genome-wide association studies.

			Exposure (PCSK9 Levels)			Outcome (Lipoprotein(a) Levels)			Two Sample IV Analysis		
Instrumental SNP	Gene	Effect Allele/Other Allele	β	SE	*p* Value	β	SE	*p* Value	β	SE	*p* Value
rs11206510	*PCSK9*	T/C	−0.051	0.009	8.98 × 10^−8^	−0.0035	0.0034	0.315	0.068	0.068	0.31493
rs11591147	*PCSK9*	G/T	−0.315	0.037	1.94 × 10^−17^	−0.0366	0.0102	3 × 10^−4^	0.116	0.032	0.000339
rs2479409	*PCSK9*	G/A	−0.052	0.009	7.41 × 10^−9^	−0.0031	0.0028	0.274	0.060	0.054	0.273589
rs45448095	*PCSK9*	C/T	−0.085	0.012	4.40 × 10^−12^	0.0003	0.0038	0.928	−0.004	0.045	0.92752
rs499718	*PCSK9*	T/C	0.062	0.012	7.81 × 10^−8^	0.0026	0.0035	0.453	0.042	0.056	0.453332
rs557435	*PCSK9*	A/G	0.050	0.010	2.22 × 10^−7^	0.0043	0.0034	0.209	0.087	0.069	0.209168

SNP: single nucleotide polymorphism.

**Table 5 ijms-24-14668-t005:** Mendelian randomization estimates from each method to access the causal effect of plasma PCSK9 levels on Lp(a) concentrations.

MR Method	Number of SNPs	β	SE	Association *p* Value	Cochran’s Q Statistic	Heterogeneity *p* Value
Inverse-variance weighted	6	0.06933	0.02007	0.0005528	5.079	0.4063
MR-Egger	6	0.1083	0.0383	0.04741	3.658	0.4063
Weighted median	6	0.06628	0.02888	0.02176	NA	NA

MR: Mendelian randomization; SE: standard error; SNP: single nucleotide polymorphism.

## Data Availability

The data presented in this study are available upon request from the corresponding author.

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
