# Peer review of "Exploring PCSK9 Genetic Impact on Lipoprotein(a) via Dual Approaches: Association and Mendelian Randomization"

_ijms, 2023, doi:10.3390/ijms241914668_

Round 1

Reviewer 1 Report

Chang YC et al. and the authors have described the causal relationship between PCSK9 E670G and Lp(a) in a small population. The results are well-represented.   Minor comments:  

Line 51: Kindly mCould you the percentage of Lp(a) reduction in patients treated with PCSK9 inhibitor.

Line 53: "Alterations in PCKS9 function". Clarify the kind of alterations seen in carriers of PCKS9 E670G.

The work is well described. Minor editing required

Author Response

  1. Response: We greatly appreciate the recommendations of the reviewer. We have added the percentage of Lp(a) reduction in patients treated with PCSK9 inhibitor in the revised version.
  2. Response: We have added the details regarding the alterations seen in carriers of PCSK9 E670G. Both gain of loss of function have been reported in previous studies.

Reviewer 2 Report

The manuscript presented by the authors suggests that there is an association between six SNP-type polymorphisms of the PCSK9 gene and low levels of Lipoprotein using an MRI model, using a study group of 614 individuals, and the statistics obtained from two previous GWAS, however,  

1. It is not clear how the six GWAS SNPs mentioned in this study were selected, so it is suggested to clarify what was the criterion for choosing this group of six SNPs used in this study, in addition to the selection of the p-value threshold and LD R square values below 0.5.

2. In relation to the group of 614 individuals, clarify how the statistical power of these 614 individuals used in this study was calculated.

3. It is suggested to include in the results section the interpretation of the ORs in Table 3 and the beta values in Table 5.

4. The authors are suggested to comment and evaluate the statistical power in the Conclusion section of this study.

Author Response

  1. Response: We greatly appreciate the recommendations of the reviewer. We have made an adjustment in our text “2.4 MR study” section to enhance the clearness of the criteria for choosing these six SNPs as instrumental variables as follow: Previous genome-wide association studies (GWAS) consistently highlighted the PCSK9 gene locus as strongly associated with plasma PCSK9 levels. Accordingly, we meticulously selected six independent single nucleotide polymorphisms (SNPs) within the PCSK9 gene as instrumental variables (IVs). Our SNP choices adhered to stringent criteria, including significant associations with PCSK9 levels that had been consistently replicated in two independent GWAS studies. These associations yielded p-values reaching or exceeding the genome-wide significance threshold of 5.00 × 10–7. Moreover, we ensured that the pairwise linkage disequilibrium (LD) r2 values between these SNPs were all below 0.5, as outlined in the methods section.
  2. Response: Thank you for this important comment. Since we did not conduct a power analysis before the study to determine the required sample size, we performed a post hoc power analysis with G*Power software based on the results obtained in our study to address the reviewer’s concern. We included the results of post hoc statistical power and the methods of analysis in the revised Discussion section, mentioning its implication of a small sample size with inadequate power in our study as a major limitation, as follows: First, the genetic association study involves a relatively small sample size. The post hoc statistical power analysis, assuming an odds ratio of approximately 0.5 for G allele carriers in terms of a 38% prevalence of average-risk or a 19% prevalence of high-risk hyperlipoproteinemia(a), indicates that this study's statistical power, given the sample size of 614 participants and the 11.9% incidence of AG and GG genotypes, is only 68% and 40%, respectively, at a significance level of 0.05. While our study provides valuable insights, it is clear that a larger sample size would be essential to validate and strengthen our findings.
  3. Response: According to the reviewer’s suggestion, we have included the interpretation of the ORs in Table 3 and the beta values in Table 5 as follows: “These results suggest that individuals with the G allele have a significantly lower risk of developing hyperlipoproteinemia(a), with a 50% lower chance for those at average risk and a 60% lower chance for those at high risk, compared to individuals with the AA genotype.” And “Specifically, in the IVW analysis, each unit increase in plasma PCSK9 levels corresponds to a 0.06933 unit increase in irnt Lp(a) levels (SE = 0.02007, p = 0.0005528).”
  4. Response: According to the reviewer’s suggestion, we have evaluated and shown the statistical power in the revised Discussion and comment on it with a larger population is necessary to enhance the robustness of our findings in the revised Conclusion section.

Reviewer 3 Report

This original paper dedicated to reveal a causal relationship between PCSK9 levels and Lp(a) concentrations. Authors included 614 healthy subjects during routine health examinations, 19% of individuals exhibit levels greater than 30 mg/dL.

1)    In this subgroup TC and LDL-C levels were higher, and this is well known and showed before.

2)    Carriers of the PCSK9 670G allele (AG or GG) had notably lower LDL-C levels (107 ± 32 mg/dL) compared to AA homozygous subjects (117 ± 33 mg/dL) and diminished hyperlipoproteinemia(a) frequencies.

3)    To explore the potential causal relationship between plasma PCSK9 and Lp(a) levels, they conducted a two-sample mendelian randomisation study. They selected six independent single nucleotide polymorphisms (SNPs) within the PCSK9 gene as instrumental variables. In this special analysis they found a causal relationship between PCSK9 levels and Lp(a) concentrations.

The paper is well written and illustrated, the results are discussed thoroughly.

Minor:
1. The abstract is not structured and does not reflect the paper results exactly.

Major:

1.    This is a small study. It is not clear how PCSK9 gene can control Lp(a) concentration. Probably, the relationship is epigenetic. It is not possible to generalize or extrapolate study results into clinical science and practice. The larger study is needed.

Author Response

  1. Response: We greatly appreciate the recommendations of the reviewer. We followed the instruction of the Journal: The abstract should be a single paragraph and should follow the style of structured abstracts, but without headings (Background, Methods, Results and Conclusion). We have made a slight adjustment to enhance the clearness as follows: [Backgrounds:] Previous investigations have suggested an association between the PCSK9 common polymorphism E670G and Lipoprotein(a) (Lp(a)) levels, as well as a link between plasma PCSK9 levels and Lp(a) concentrations. However, the causal relationship between plasma PCSK9 and Lp(a) levels remains uncertain. [Methods:] In this study, we explored the association between the PCSK9 E670G polymorphism and Lp(a) levels in 614 healthy Taiwanese individuals. Employing a two-sample Mendelian Randomization (MR) analysis using openly accessible PCSK9 and Lp(a) summary statistics from the genome-wide association studies (GWAS) and UK Biobank, we aimed to determine if a causal link exists between plasma PCSK9 levels and Lp(a) concentrations. [Results:] Our findings reveal that the E670G G allele is independently associated with a decreased likelihood of developing elevated Lp(a) levels. This association persists even after adjusting for common cardiovascular risk factors and irrespective of lipid profile variations. The MR analysis, utilizing six PCSK9 GWAS-associated variants as instrumental variables to predict plasma PCSK9 levels, provides compelling evidence of a causal relationship between plasma PCSK9 levels and Lp(a) concentration. [Conclusions:] In conclusion, our study not only replicates the association between the PCSK9 E670G polymorphism and Lp(a) levels but also confirms a causative relationship between PCSK9 levels and Lp(a) concentrations through MR analysis.
  2. Response: Thank you this important comment. Although, we have mentioned the implication of our study results on clinical science and practice as reinforcing the potential of PCSK9 inhibitors to reduce Lp(a) levels effectively. To address this concern, we have mentioned this limitation in the revised Discussion and Conclusion as follows: “Lastly, the absence of mechanistic investigations to elucidate the underlying pathways and mechanisms by which the PCSK9 gene controls Lp(a) levels is a notable limitation. As a result, caution should be exercised when attempting to generalize or extrapolate our study results to clinical science and practice.” And “However, it is crucial to emphasize that validation from a larger, more diverse population and mechanistic investigations are necessary to enhance the robustness of our findings.”

Reviewer 4 Report

Authors: Ya-Ching Chang, Lung-an Hsu and Yu-lin Ko

Manuscript: Exploring PCSK9 genetic impact on Lp(a) via dual approaches: association and mendelian randomization

Critics:  This is a well written manuscript.  Even though the sample number is small and limited to one ethnic background.  It did provide some insights on PCSK9 E670G on PCSK9 levels and Lp(a) concentrations.

Mendelian randomization is an important analysis used by the authors on this study, the authors should provide a paragraph explanation on this analysis for readers to appreciate the significant of this analysis.

Author Response

Response: We greatly appreciate the recommendations of the reviewer. We have added a paragraph explanation on MR analysis in the revised Introduction as follow: MR is a crucial analytical approach, offering a robust framework to investigate causal relationships between specific genetic variants, biomarkers, and clinical outcomes. MR leverages the principles of Mendel's laws of inheritance, utilizing genetic variants as instrumental variables to mimic a randomized controlled trial setting. This allows us to explore the causal effects of exposures such as plasma PCSK9 level on clinical outcomes like Lp(a) concentrations using PCSK9 genetic variants as proxies. By relying on naturally occurring genetic variation, MR mitigates issues of reverse causality and confounding that often challenge traditional observational studies.

Round 2

Reviewer 3 Report

Authors responded to all queries appropriately and recognize important study limitations and unmet need for larger studies in this area.

Though, I am sure that these data are not sufficient to establish causal relationship between PCSK9 polymorphism and Lp(a) levels, I assume now that these findings sound well and better than before and the paper may be published after the Editor final decision taking into account all study defects.